# Sex-specific cardiac remodeling in early and advanced stages of hypertrophic cardiomyopathy

**Louise L. A. M. Nijenkamp**[1]*, **Ilse A. E. Bollen**[1], **Hans W. M. Niessen**[2], **Cris G. dos Remedios**[3], **Michelle Michels**[4], **Corrado Poggesi**[5], **Carolyn Y. Ho**[6], **Diederik W. D. Kuster**[1,7], **Jolanda van der Velden**[1,7]

**1** Physiology, Amsterdam UMC, Vrije Universiteit Amsterdam, Amsterdam Cardiovascular Sciences, Amsterdam, The Netherlands, **2** Pathology and Cardiac Surgery, Amsterdam UMC, Vrije Universiteit Amsterdam, Amsterdam Cardiovascular Sciences, Amsterdam, The Netherlands, **3** Muscle Research Unit, Bosch Institute, University Sydney, Sydney, Australia, **4** Department of Cardiology, Erasmus Medical Center, Rotterdam, The Netherlands, **5** Dipartimento di Medicina Sperimentale e Clinica, Università degli Studi di Firenze, Firenze, Italy, **6** Brigham and Women's Hospital, Harvard Medical School, Boston, MA, United States of America, **7** Netherlands Heart Institute, Utrecht, the Netherlands

* la.nijenkamp@amsterdamumc.nl

**Data Availability Statement:** All relevant data are within the paper and its Supporting Information files.

## Abstract

Hypertrophic cardiomyopathy (HCM) is the most frequent genetic cardiac disease with a prevalence of 1:500 to 1:200. While most patients show obstructive HCM and a relatively stable clinical phenotype (stage II), a small group of patients progresses to end-stage HCM (stage IV) within a relatively brief period. Previous research has shown sex-differences in stage II HCM with more diastolic dysfunction in female than in male patients. Moreover, female patients more often show progression to heart failure. Here we investigated if differences in functional and structural properties of the heart may underlie sex-differences in disease progression from stage II to stage IV HCM. Cardiac tissue from stage II and IV patients was obtained during myectomy (n = 54) and heart transplantation (n = 10), respectively. Isometric force was measured in membrane-permeabilized cardiomyocytes to define active and passive myofilament force development. Titin isoform composition was assessed using gel electrophoresis, and the amount of fibrosis and capillary density were determined with histology. In accordance with disease stage-dependent adverse cardiac remodeling end-stage patients showed a thinner interventricular septal wall and larger left ventricular and atrial diameters compared to stage II patients. Cardiomyocyte contractile properties and fibrosis were comparable between stage II and IV, while capillary density was significantly lower in stage IV compared to stage II. Women showed more adverse cellular remodeling compared to men at stage II, evident from more compliant titin, more fibrosis and lower capillary density. However, the disease stage-dependent reduction in capillary density was largest in men. In conclusion, the more severe cellular remodeling in female compared to male stage II patients suggests a more advanced disease stage at the time of myectomy in women. Changes in cardiomyocyte contractile properties do not explain the progression of stage II to stage IV, while reduced capillary density may underlie disease progression to end-stage heart failure.

**Funding:** Grant support received by Jolanda van der Velden: The study was sponsored by the Netherlands Cardiovascular Research Initiative, an initiative with support of the Dutch Heart Foundation, CVON2011-11 ARENA and CVON2014-40 DOSIS. The funders had no role in study design, data collection and analysis, decision to publish, or preparation of the manuscript. No relations with industry.

**Competing interests:** The authors have declared that no competing interests exist.

## Introduction

Hypertrophic cardiomyopathy (HCM) is the most prevalent genetic cardiac disease occurring in 2–5 per 1000 individuals, and is caused by mutations in genes encoding sarcomeric proteins.[1–3] The defining feature of HCM is unexplained left ventricular (LV) hypertrophy that mainly affects the interventricular septum (IVS). In addition to hypertrophy, the diseased myocardium is characterized by increased interstitial fibrosis, myofibrillar and cardiomyocyte disarray and vascular abnormalities.[4,5] Clinical symptoms generally appear between 20–50 years of age and can range from shortness of breath to atrial fibrillation. In the majority of patients, these symptoms can be managed with therapy, and individuals have a normal life expectancy.[6] However, a subset of patients suffers from life-threatening complications such as sudden cardiac arrest at a young age or progresses to end-stage heart failure.[6]

To classify the different forms of HCM, *Olivotto et al.*[7] described different stages in cardiac disease progression ranging from unaffected mutation carriers to end-stage failing HCM patients.[8] The majority of HCM patients develop a 'classic' (stage II) form of HCM with the characteristic septal thickening, LV outflow tract obstruction (LVOTO) and diastolic dysfunction. Approximately 5–10% of all HCM patients progress to the severe end-stage of HCM (stage IV), which is characterized by thinning of the IVS and LV wall, and left atrial dilation, extensive fibrosis and impaired systolic function. These patients show a decrease in NYHA classification and are often diagnosed with or develop atrial fibrillation.[9] The progression from the 'classic' stage of HCM, via adverse remodeling (stage III) to stage IV shows a relatively short clinical course of approximately 6.5 years.[10] It is unclear which factors underlie the transition from a stable stage II to stage IV of HCM in a relatively small but severely ill group of patients.

Notably, the lifetime risk of heart failure is higher in women than in men.[11] At first evaluation, female HCM patients are older and present more often with symptoms.[12,13] Moreover, women show a higher risk of progression to heart failure (stage IV).[12,14] We recently showed more severe diastolic dysfunction in female compared to male HCM patients at the time of myectomy (stage II), which coincided with more advanced tissue remodeling in women compared to men.[15] Here we investigated if differences in functional and structural properties of the heart may underlie sex-differences in disease progression from stage II to IV of HCM. Functional measurements in single cardiomyocytes were combined with analyses of protein expression, fibrosis and capillary density.

## Methods

### Myocardial samples

Cardiac samples were collected from 54 patients with obstructive HCM (stage II) (44% female) carrying a *MYH7* (myosin heavy chain; n = 14) or a *MYBPC3* (myosin-binding protein-C; n = 40) mutation (stage II) and 10 end-stage (stage IV) patients (40% female) carrying a *MYH7* (n = 6), a *MYBPC3* (n = 3) or a *TNNT2* (troponin T; n = 1) mutation. Fig 1A illustrates the different gene mutations and Fig 1B shows where mutations are located in the protein. Table 1 provides an overview of the gene mutations of all patients. Patient samples were compared with non-failing control samples (n = 30; 40±14 years; 43% female; S1 Table) without a history of cardiac abnormalities. IVS tissue from the stage II HCM group was collected during myectomy to relieve LVOTO. Cardiac tissue from stage IV HCM was obtained during heart transplantation surgery and consisted of IVS (n = 6) and LV tissue (n = 4). Control samples included IVS (n = 3) and LV (n = 27) tissue. All samples were immediately frozen and stored in liquid nitrogen. This study was approved by the local ethics board of the Erasmus Medical

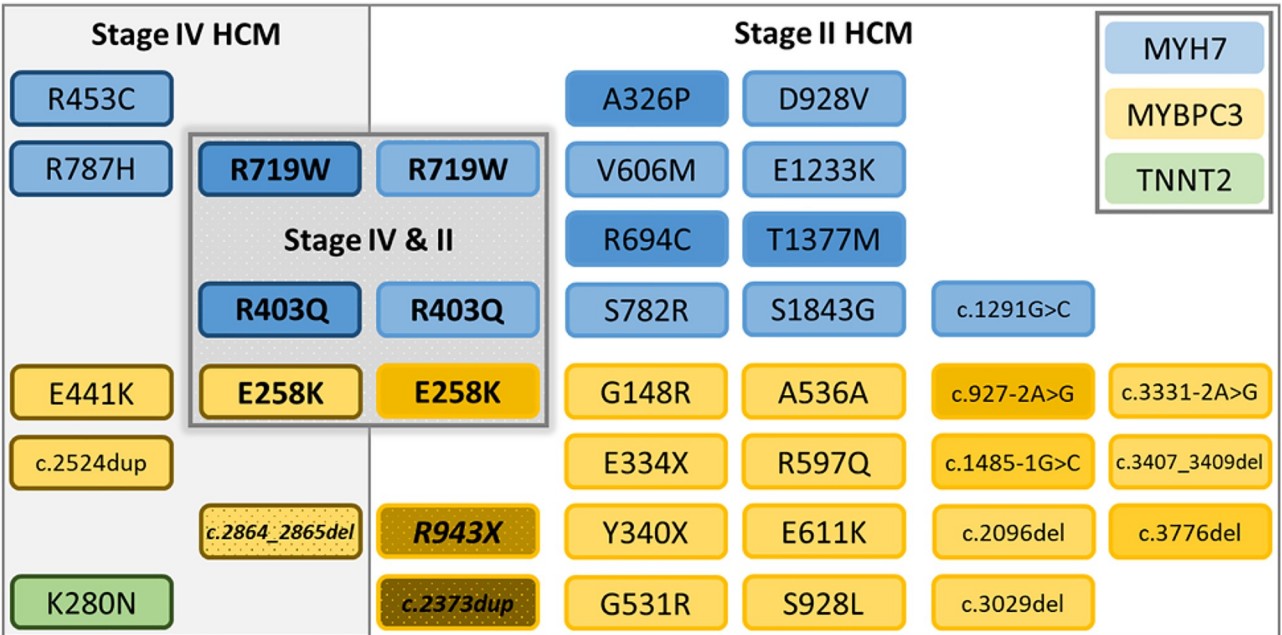

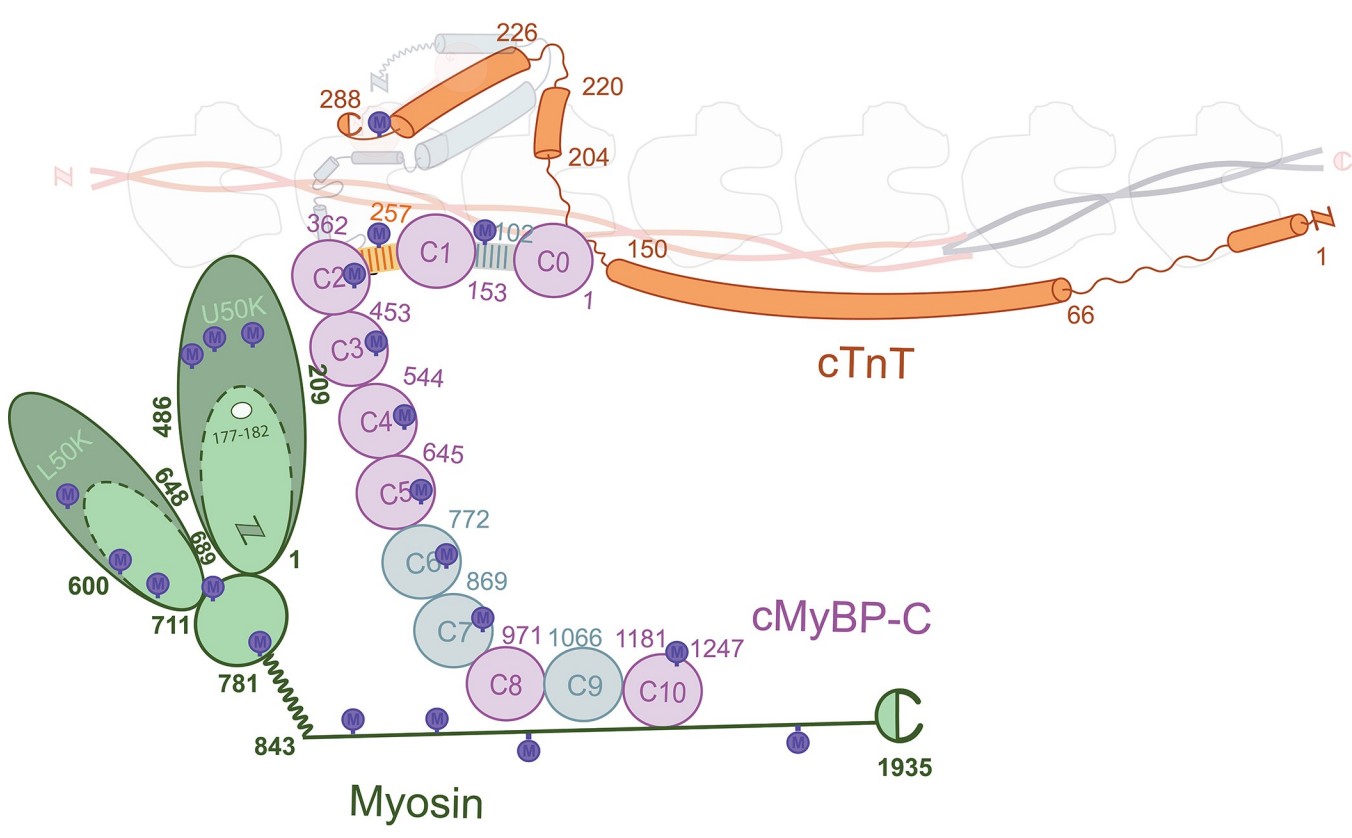

**Fig 1.** A. Mutations present in the patient samples. Mutations of patients that were included in this study. Myosin heavy chain gene (*MYH7*) mutations are depicted in blue squares of which two mutations (R719W and R403Q) were present in both stage II and IV (end-stage) HCM patient groups. Myosin-binding protein–C gene (*MYBPC3*) mutations are depicted in the yellow squares of which one mutation (E258K) was present in both HCM patient groups. The three Dutch founder mutations are depicted in the dotted squares. The troponin T gene (*TNNT2*) mutation is depicted in the green square and was only present in the end-stage patient group. Color tone indicates the number of patients carrying the specific mutation, with the darkest tone representing most patients (patient details are given in Table 1). B. Mutation location. Schematic of 3 main HCM sarcomere proteins: myosin heavy chain in green (Myosin), cardiac myosin-binding protein-C in purple (cMyBP-C) and cardiac troponin T in orange (cTnT). The location of the mutations is indicated with the blue circles (M). The letters N and C stand for the N-terminus and C-terminus respectively. The numbers indicate the amino acids of the sarcomere proteins.

Center (protocol number MEC-2010-40) and written informed consent of patients was obtained. Non-failing donor samples were acquired from the University of Sydney, Australia, with the ethical approval of the Human Research Ethics Committee (#2012/2814). We acknowledge the uneven distribution between patient numbers and mutations and tissue locations (IVS and LV). However, our tissue characterization was dependent on available cardiac tissue and clinical parameters. Due to limited tissue availability, not all analyses could be performed in all patient/control samples.

## Echocardiographic measurements

Echocardiographic studies were done with commercially available systems and analyzed according to the American Society of Echocardiography guidelines.[16] Maximal wall thickness, left atrial diameter (LAD), LV end-diastolic diameter (LVEDD), and LVOTO gradient were measured. LVOTO was defined as a gradient $\geq 30$ mmHg at rest or during provocation. Mitral valve inflow was recorded using pulsed wave Doppler from the apical four chamber view. Mitral E and A velocity (cm/s) and deceleration time (ms) were measured. Pulsed wave tissue Doppler imaging was used to measure septal e' velocity (cm/s). Continuous wave Doppler in the parasternal and apical four chamber was used to measure tricuspid regurgitation (TR) velocity (m/s). Echocardiographic data and medication are shown in Table 1. For the end-stage HCM group, a limited set of echocardiographic data was obtained, and not all parameters were obtained for stage II patients.

Diastolic dysfunction was graded as follows: grade I when E/A ratio $\leq 0.8$ and E peak velocity $\leq 50$ cm/s; grade III when E/A ratio $\geq 2$. In patients with E/A ratio $\leq 0.8$ and E peak velocity $> 50$ cm/s or E/A ratio $> 0.8$ but $< 2$, the E/e' ratio ($>14$), LADi ($>24$) and TR velocity ($> 2.8$ m/s) were used to further differentiate diastolic function. When $\geq 2$ out of 3 variables were abnormal, LA pressure was elevated and grade II diastolic dysfunction was present. When 1 out of 3 variables was abnormal, grade I diastolic dysfunction was present.[17]

## Isometric force measurements

Force measurements were performed in mechanically isolated single, membrane-permeabilized, cardiomyocytes as described previously.[18,19] In short, we measured passive tension at a range of sarcomere lengths (SL)(1.8–2.4 μm). All passive forces were normalized to cardiomyocyte cross-sectional area (CSA) (i.e. CSA = width x depth x π/4). Active tension (maximal force) was measured at SL 2.2 μm, and we determined myofilament calcium sensitivity ($EC_{50}$) by activating cardiomyocytes in solutions with different calcium concentrations.

## Protein analyses

Titin isoform gel electrophoresis was performed as previously described.[19,20] Samples were measured in triplicate, of which the mean was used.

**Table 1. Patient information and gene mutations.**

| Sample code | SEX | AGE (Y) | GENE | MUTATION CODE | DOMAIN & INFO | IVS (mm) | IVSi | LAD | LADi | LVEDD | LVEDDi | LVOTO | E/A RATIO | E WAVE | TR VELOCITY | DRUG REGIMEN |
|---|---|---|---|---|---|---|---|---|---|---|---|---|---|---|---|---|
| **STAGE II** | | | | | | | | | | | | | | | | |
| HCM 2 | F | 32 | *MYBPC3* | Y842L | C6 / DF | 30 | 20 | - | - | 40 | 27 | - | - | - | 0.3 | Bb |
| HCM 3 | F | 39 | *MYPBC3* | Y842L | C6 / DF | 20 | 11 | 40 | 23 | 38 | 21 | 49 | - | 0.8 | - | Ccb, diuretics |
| HCM 4 | F | 44 | *MYBPC3* | Y842L | C6 / DF | 20 | 13 | 45 | 30 | 40 | 27 | 94 | - | 0.8 | - | Ccb |
| HCM 5 | F | 41 | *MYBPC3* | c.1458-1G>C | Intr 16 | 22 | 13 | 46 | 27 | 37 | 21 | 92 | 3.8 | 1.1 | - | Bb, ccb |
| HCM 7 | F | 44 | *MYBPC3* | Y842L | C6 / DF | 17 | 11 | 40 | 25 | 42 | 26 | | - | - | - | Ccb |
| HCM 26 | F | 57 | *MYBPC3* | Y842L | C6 / DF | 24 | 11 | 62 | 28 | 41 | 18 | 74 | 0.7 | 0.4 | - | Bb |
| HCM 34 | F | 47 | *MYBPC3* | R597Q | C4 | 20 | 10 | 46 | 23 | 42 | 21 | 38 | 2 | 0.9 | 3 | Bb, ccb, statin |
| HCM 35 | F | 65 | *MYBPC3* | A536A | C3 | 19 | - | - | - | 45 | - | 18 | - | - | 2.7 | Bb, ccb |
| HCM 52 | F | 24 | *MYBPC3* | R943x | C7 / DF | 24 | 14 | 44 | 26 | 35 | 20 | 34 | 1.12 | 0.9 | 2.5 | Bb |
| HCM 60 | F | 45 | *MYBPC3* | c.3029delA | C8 | 18 | 9 | 46 | 24 | 42 | 22 | 125 | 0.68 | 0.6 | 1.9 | Bb, ccb |
| HCM 94 | F | 66 | *MYBPC3* | E611K | C4 | 15 | 8 | 53 | 27 | 43 | 22 | 55 | - | - | 2 | Bb, ccb, ACE, statin, oac |
| HCM 113 | F | 21 | *MYBPC3* | R943x | C7 / DF | 45 | 28 | 43 | 26 | 40 | 25 | 27 | 1.26 | 0.7 | 2.3 | Bb, ccb |
| HCM 116 | F | 53 | *MYBPC3* | R943x | C7 / DF | - | - | - | - | - | - | - | - | - | - | - |
| HCM 121 | F | 54 | *MYBPC3* | Q1259R | C10 | 20 | 12 | 33 | 21 | 37 | 23 | 45 | 0.93 | 1 | 1.5 | Ccb |
| HCM 123 | F | 59 | *MYBPC3* | Y842L | C6 / DF | 21 | - | 45 | - | 55 | - | 64 | 1.2 | 1.4 | 3.1 | Bb, ccb, oac, ACE, Diuretics |
| HCM 180 | F | 51 | *MYBPC3* | Y340x | MM | 32 | - | - | - | 45 | - | 70 | 2.25 | 0.9 | - | Bb |
| HCM 148 | F | 51 | *MYBPC3* | G531R | C3 | 20 | 9 | 58 | 26 | 51 | 23 | 56 | 0.97 | 0.7 | - | - |
| HCM 150 | F | 57 | *MYBPC3* | P699Q | C5 | 24 | 12 | 49 | 25 | 50 | 26 | 68 | 2.5 | 1.3 | - | - |
| HCM 12 | M | 37 | *MYBPC3* | c.927-2A>G | Intr 11 | 19 | 9 | 41 | 20 | 42 | 21 | 44 | 2 | 1 | - | Bb |
| HCM 33 | M | 48 | *MYBPC3* | c.927-2A>G | Intr 11 | - | - | - | - | - | - | 82 | - | - | - | Bb |
| HCM 36 | M | 22 | *MYBPC3* | c.927-2A>G | Intr 11 | 30 | - | 60 | - | 44 | - | 71 | 0.73 | 0.4 | - | Bb, ccb |
| HCM 42 | M | 32 | *MYBPC3* | Y842L | C6 / DF | 23 | 13 | 47 | 26 | 43 | 24 | 64 | 1.03 | 0.7 | - | Bb, ccb |
| HCM 43 | M | 60 | *MYBPC3* | Y842L | C6 / DF | 23 | - | 52 | - | 45 | - | 77 | 2 | 0.8 | - | Bb, ccb |
| HCM 47 | M | 55 | *MYBPC3* | Y1136d | C9 | 25 | 13 | - | - | 40 | 21 | 96 | 1.57 | 1 | 2.9 | Ccb |
| HCM 62 | M | 36 | *MYBPC3* | E258K | MM | 27 | 14 | 37 | 19 | 32 | 16 | - | 1.21 | 0.4 | 1.8 | Bb, ACE, diuretics |
| HCM 63 | M | 33 | *MYBPC3* | R943x | C7 / DF | 21 | 12 | 40 | 22 | 46 | 26 | 25 | 2.38 | 1.1 | - | Bb, ccb |
| HCM 66 | M | 45 | *MYBPC3* | Q1259R | C10 | - | - | - | - | - | - | - | - | - | - | - |
| HCM 71 | M | 49 | *MYBPC3* | R943x | C7 / DF | 16 | 7 | 47 | 22 | 44 | 20 | - | 0.78 | 0.5 | 2.3 | Bb |
| HCM 82 | M | 71 | *MYBPC3* | S928L | C7 | 20 | 11 | 52 | 29 | 49 | 28 | 64 | 0.87 | 0.7 | 2.2 | Bb, oac |
| HCM 83 | M | 17 | *MYBPC3* | Y842L | C6 / DF | - | - | - | - | - | - | - | - | - | - | - |
| HCM 101 | M | 20 | *MYBPC3* | c.1458-1G>C | Intr 16 | 38 | 17 | 42 | 19 | - | - | - | 1.25 | 0.5 | - | Bb |
| HCM 103 | M | 26 | *MYBPC3* | Y842L | C6 / DF | 20 | 9 | 47 | 22 | 47 | 22 | 13 | 1.33 | 0.7 | 1.4 | Bb |
| HCM 104 | M | 33 | *MYBPC3* | Y842L | C6 / DF | 24 | 13 | 33 | 18 | 40 | 22 | 31 | 1.08 | 0.6 | - | Bb |
| HCM 110 | M | 39 | *MYBPC3* | E334x | MM | 18 | 8 | - | - | 46 | 21 | 9 | 2.48 | 0.8 | 2.4 | Ccb |
| HCM 120 | M | 27 | *MYBPC3* | Y842L | C6 / DF | 24 | 13 | 39 | 21 | 37 | 20 | 61 | 1.42 | 0.8 | 2.3 | Bb |
| HCM 122 | M | 50 | *MYBPC3* | c.3331-2A>G | Intr 30 | 22 | 10 | - | - | 45 | 21 | 58 | 0.67 | 0.7 | 2.5 | Bb, ccb |
| HCM 124 | M | 53 | *MYBPC3* | R943x | C7 / DF | 21 | 10 | 43 | 21 | 47 | 23 | 41 | 0.64 | 0.5 | 2.2 | Bb |
| HCM 133 | M | 58 | *MYBPC3* | G148R | PA | 21 | 10 | 44 | 21 | 40 | 19 | 19 | 0.86 | 0.6 | 2.4 | Bb, statin, oac |
| HCM 141 | M | 46 | *MYBPC3* | E258K | MM | 23 | - | 51 | - | 51 | - | 20 | 1.22 | 0.7 | - | Bb |
| HCM 149 | M | 46 | *MYBPC3* | E258K | MM | 26 | - | - | - | - | - | 72 | - | - | - | - |
| HCM 27 | F | 58 | *MYH7* | T1377M | MT | 20 | 13 | 48 | 30 | - | - | 100 | - | 0.6 | 2.8 | Bb |
| HCM 42B | F | 46 | *MYH7* | V606M | MH | 20 | 9 | 51 | 24 | - | - | 77 | 3.38 | 0.8 | 2.6 | Bb, ccb |
| HCM 157 | F | 65 | *MYH7* | S1843C | MT | 22 | 10 | 58 | 27 | 42 | 20 | - | 2.37 | 1.9 | 3.3 | Ccb, statin |
| HCM 166 | F | 66 | *MYH7* | R694C | MH | 16 | - | 41 | - | - | - | 41 | 0.79 | 1.1 | - | Bb, ccb, ASA |
| HCM 144 | F | 30 | *MYH7* | S782R | Hinge | 29 | - | - | - | 42 | - | 128 | - | - | - | - |
| HCM 154 | F | 6 | *MYH7* | R719W | MH / Fam I | 20 | 26 | - | - | - | - | 46 | - | - | - | - |
| HCM 32 | M | 43 | *MYH7* | T1377M | MT | 21 | 9 | 51 | 21 | 47 | 19 | 121 | 1 | 0.6 | - | Bb, diuretics |
| HCM 80 | M | 34 | *MYH7* | c.1291G>C | MT | 17 | 8 | - | - | 45 | 21 | 85 | 1.45 | 0.9 | 1.4 | Bb, ccb |

(*Continued*)

**Table 1.** (Continued)

| Sample code | SEX | AGE (Y) | GENE | MUTATION CODE | DOMAIN & INFO | IVS (mm) | IVSi | LAD | LADi | LVEDD | LVEDDi | LVOTO | E/A RATIO | E WAVE | TR VELOCITY | DRUG REGIMEN |
|---|---|---|---|---|---|---|---|---|---|---|---|---|---|---|---|---|
| HCM 106 | M | 35 | MYH7 | D928V | MT | 16 | 8 | - | - | 47 | 24 | 16 | 0.93 | 0.8 | 2.1 | Bb |
| HCM 114 | M | 69 | MYH7 | A326P | MH | 19 | - | 43 | - | 33 | - | 71 | 0.6 | 0.6 | 2.1 | Bb, statin, ASA |
| HCM 119 | M | 41 | MYH7 | E1233K | MT | 20 | 9 | 56 | 26 | - | - | 81 | 1.26 | 1.1 | 1.4 | Bb |
| HCM 130 | M | 45 | MYH7 | A326P | MH | 18 | 9 | 42 | 21 | 50 | 25 | 27 | 2.6 | 1.3 | 3.3 | Bb, statin, noac |
| HCM 145 | M | 28 | MYH7 | R694C | MH | 42 | - | 52 | - | 38 | - | 30 | 1.5 | 0.6 | - | Bb |
| HCM 143 | M | 26 | MYH7 | R403Q | MH | 34 | - | - | - | 50 | - | 85 | - | - | - | - |
| MEAN STAGE II | | | | | | 23±6 | 12±4 | 47±7 | 24±3 | 43±5 | 22±3 | 59±30 | 1.75±0.5 | 1±0 | 2.5±0.7 | |
| STAGE IV | | | | | | | | | | | | | | | | |
| HCM 147 | F | 36 | MYBPC3 | P955R | C7 / DF | - | - | - | - | 75 | - | - | - | - | - | - |
| HCM 142 | M | 46 | MYPBC3 | E258K + E441K | MM + C2 | 18 | - | 51 | - | 50 | - | 5 | - | - | - | Bb, ACE, diuretics |
| HCM 137 | M | 54 | MYBPC3 | Y842L | C6 | - | - | - | - | 60 | - | - | - | - | - | - |
| HCM 183 | F | 39 | MYH7 | R453C | MH | - | - | - | - | 54 | 31 | 10 | - | - | - | - |
| HCM 151 | F | 59 | MYH7 | R403Q | MH | - | - | 70 | 50 | 53 | 38 | 10 | - | - | - | - |
| HCM 181 | F | 40 | MYH7 | R719W | MH / Fam I | 18 | 10 | 48 | 27 | 54 | 31 | - | - | - | - | - |
| HCM 152 | M | 35 | MYH7 | R403Q | MH | - | - | - | - | - | - | - | - | - | - | - |
| HCM 153 | M | 49 | MYH7 | R719W | MH / Fam I | 16 | 9 | 52 | 29 | 50 | 28 | - | - | - | - | - |
| HCM 139 | M | 61 | MYH7 | R787H | Hinge | - | - | - | - | 58 | - | - | - | - | - | - |
| HCM 140A | M | 26 | TNNT2 | K280N | C-terminus | - | - | - | - | - | - | - | - | - | - | - |
| MEAN STAGE IV | | | | | | 17±1 | 10+1 | 55+10 | 35+13 | 57±8 | 32±4 | 8±3 | | | | |

Abbreviations: F, female; M, male; sarcomere genes *MYBPC3*, *MYH7* and *TNNT2* encoding myosin-binding protein-C, myosin heavy chain and cardiac troponin T, respectively. Domain & info: DF: Dutch founder mutation; Fam I: patient samples part of one family. Domain locations of the mutations (also illustrated in Fig 1B): MM: cMyBP-C motif (in Fig 1B depicted as yellow stripes between C1-C2); PA: Pro-Ala rich region (in Fig 1B depicted as grey stripes between C0-C1); MT: Myosin tail (in Fig 1B the dark green line); MH: Myosin head; Hinge: hinge region of myosin (in Fig 1B depicted as the curled green line); IVS, interventricular septum; IVSi, IVS indexed by body surface area (BSA); LAD, left atrial diameter; LADi, LAD indexed by BSA; LVEDD, left ventricular end-diastolic diameter; LVEDDi indexed by BSA; LVOTO, left ventricular outflow tract obstruction; E/A ratio, ratio of mitral valve early (E) and late (A) velocity; E wave, mitral valve early velocity (cm/s); TR velocity, tricuspid regurgitation velocity (m/s); bb, betablocker; ccb, calcium channel blocker; ACE, ACE-inhibitor; (N)OAC, (novel) oral anticoagulant; ASA, antiplatelet therapy (acetylsalicylic acid).

## Histomorphometrical analyses

Cardiomyocyte myofibril density (MFD) was measured using Electron Microscopy (EM) as described previously.[21] MFD was calculated by the sum of myofibril area relative to the total cardiomyocyte area and expressed as a percentage.[21,22] To determine the extent of interstitial and replacement fibrosis, cryosections were stained using Picro-Sirius Red, and fibrosis was expressed as collagen volume fraction (CVF%). Capillary density was determined by the number of capillaries per $mm^2$ and per cardiomyocyte. Cryosections were incubated with a primary antibody (Monoclonal Mouse anti-Human CD31, Endothelial cell; Clone JC70A; DAKO; REF M0823) and secondary antibody (EnVision HRP α-mouse/rabbit (DAKO). Staining was visualized using 3.3'-diaminobenzidine (0.1 mg/mL, 0.02% $H_2O_2$), the sections were subsequently counterstained with hematoxylin.

**Table 2. Clinical characteristics of HCM stage II and IV patient groups.**

|  | stage II | stage IV | P |
|---|---|---|---|
| N (% female) | 54 (44%) | 10 (40%) |  |
| Age (years) | 43.4±14.7 | 44.5±11.3 | 0.83 |
| IVS (mm) | 22.9±6.3 | 17.3±1.2 | **<0.05** |
| LVOTO (mmHG) | 59.2±30.4 | 8.3±2.9 | **<0.001** |
| LAD (mm) | 46.5±6.9 | 55.3±10.0 | **<0.05** |
| LVEDD (mm) | 43.1±5.0 | 56.8±8.2 | **<0.0001** |

Abbreviations: IVS (interventricular septum); LAD (left atrial dimension); LVEDD (left ventricular end-diastolic dimension); LVOTO (left ventricular outflow tract obstruction).

## Data analyses

Data in figures are presented as mean±standard error of the mean per group, data in tables is presented as mean±standard deviation. If data was normally distributed means were compared with a student's T-test, a Mann-Whitney test was used when data was not normally distributed. Sex-differences were tested with a 2-way ANOVA. $P<0.05$ was considered significant, differences to controls are indicated by an asterisk (*), sex-differences are indicated by a hashtag (#).

## Results

A *MYBPC3* gene mutation was present in 43 patients (40 stage II and 3 stage IV), of which 18 were founder mutations. Most *MYH7* mutations were located in the myosin head (60%) (Fig 1B and Table 1). Average values for echocardiographic characteristics of HCM stage II and IV patients are shown in Table 2. HCM stage II and IV patients have a similar average age and male patients are dominant in both groups. All patients meet the diagnostic criteria of an IVS thickness > 15 mm (Table 1), however, the HCM stage II patients show significantly greater septal thickness than the HCM stage IV patients. This is in line with a higher LVOTO in the stage II compared to stage IV HCM. LAD and LVEDD were significantly higher in stage IV compared to stage II HCM.[23] All stage IV patients were given grade III diastolic dysfunction, while stage II patients included 29% with grade III, 29% with grade II and 42% with grade I diastolic dysfunction. Female stage II patients show more severe diastolic dysfunction compared to male patients: 86% grade II or III diastolic dysfunction in women compared to 42% grade II in men. Drug regimen was different between female and male stage II HCM patients: 80% of the men received betablockers in contrast to 58% of the women. Calcium channel blockers were prescribed more frequently to women compared to men (58% versus 27%, respectively; Table 1). Unfortunately we could only retrieve drug regimen of one of our stage IV patients.

Maximal force generating capacity (Fmax) of cardiomyocytes was significantly lower in stage II and IV HCM compared to controls (21.8±1.7 and 19.6±2.3 versus 32.8±2.9 KN/m$^2$ respectively; $p<0.01$), but did not differ between patient groups or sex (Fig 2A). Passive tension (Fpass) was lower in both stage II and IV HCM compared to controls (Fig 2B; $p<0.0001$). No sex-differences in passive tension were observed. Fig 2C shows representative images of myofibril density analyses by EM (stage II and IV HCM patient samples). Both HCM groups showed a similar decrease in MFD compared to controls ($p<0.0001$), while no sex-difference was observed (Fig 2D). Compared to controls, a significantly higher myofilament calcium-

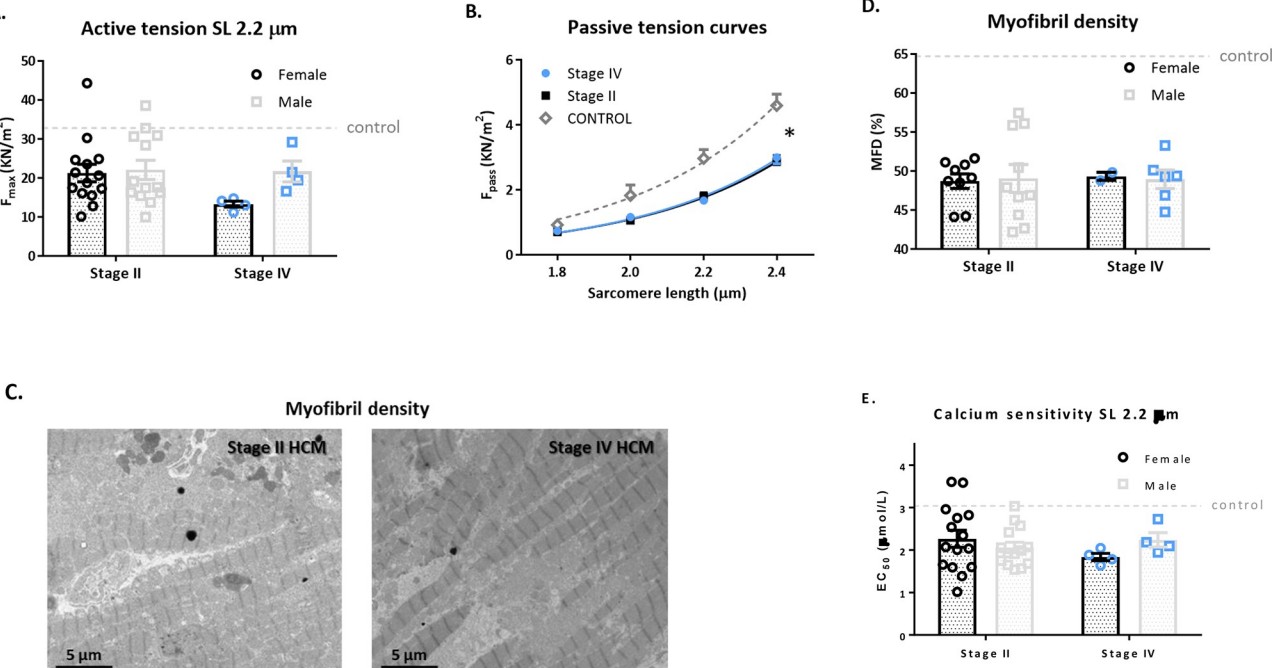

**Fig 2. Cardiomyocyte properties–no differences between stage II and IV HCM patients.** A) Maximal force development is lower in stage II (n = 26) and stage IV (n = 8) patients in comparison to controls (p<0.0001) and no sex-differences were found. B) Passive tension is lower in stage II (n = 19) and stage IV (n = 8) patients in comparison to controls (p<0.0001, no sex-differences were found). C and D) Electron Microscopy (EM) imaging was performed and showed reduced myofibril density in stage II and IV HCM patient samples compared to controls. No sex-difference was present in myofibril density. E) Myofilament calcium-sensitivity is depicted as $EC_{50}$ (the amount of calcium needed to reach 50% of maximal force). $EC_{50}$ is significantly lower in stage II and IV HCM patients compared to controls (p<0.001). No sex-differences were found. Controls used for passive tension: 1, 12–13, 20; maximal force development: 5–8, 11–12, 15–16, 18–19, 23, 26; myofilament calcium-sensitivity: 5–8, 11–12, 15–16, 18–19, 23.

sensitivity was found in both stage II and IV HCM ($EC_{50}$: 2.2±0.1 µmol/L and 3.0±0.2 versus 2.0±0.1 respectively; p<0.001), with no difference between the HCM groups or sex (Fig 2E).

The N2BA/N2B titin isoform ratio was significantly higher in both HCM groups compared to controls, without a difference between stage II and IV HCM (0.84±0.05 versus 0.81±0.12). At both HCM disease stages females show a higher N2BA/N2B ratio compared to males (Fig 3A). Analysis of fibrosis showed increased fibrosis in stage II and IV HCM patients compared to controls (4.6±0.5 and 5.5±0.9 versus 1.2±0.2% respectively; p<0.001). Representative Picro-Sirius red staining images of stage II and IV HCM are shown in Fig 3C. There was no difference between the HCM groups, however, in both disease stages women showed significantly more fibrosis than men (Fig 3B). We found significant correlations between the degree of diastolic dysfunction and titin isoform composition (Fig 3D; $R^2$: 0.13; p<0.05) and the amount of fibrosis (Fig 3E; $R^2$: 0.31; p<0.01).

Fig 4A shows images of capillary staining in a stage II and IV HCM sample. Capillary density is depicted as capillaries per $mm^2$ (Fig 4B) as well as per cardiomyocyte (Fig 4C). Capillary density is significantly lower in stage II and IV HCM patients compared to controls (Fig 4B and 4C; p<0.0001). Furthermore, both methods showed a lower capillary density in stage IV compared to stage II HCM patients (p<0.05). Capillary density in stage II HCM is significantly lower in women compared to men (p<0.05). Interestingly, the reduction in capillary density from stage II to stage IV is attributed to the reduction of capillary density in male to a similar level as observed in female HCM patients.

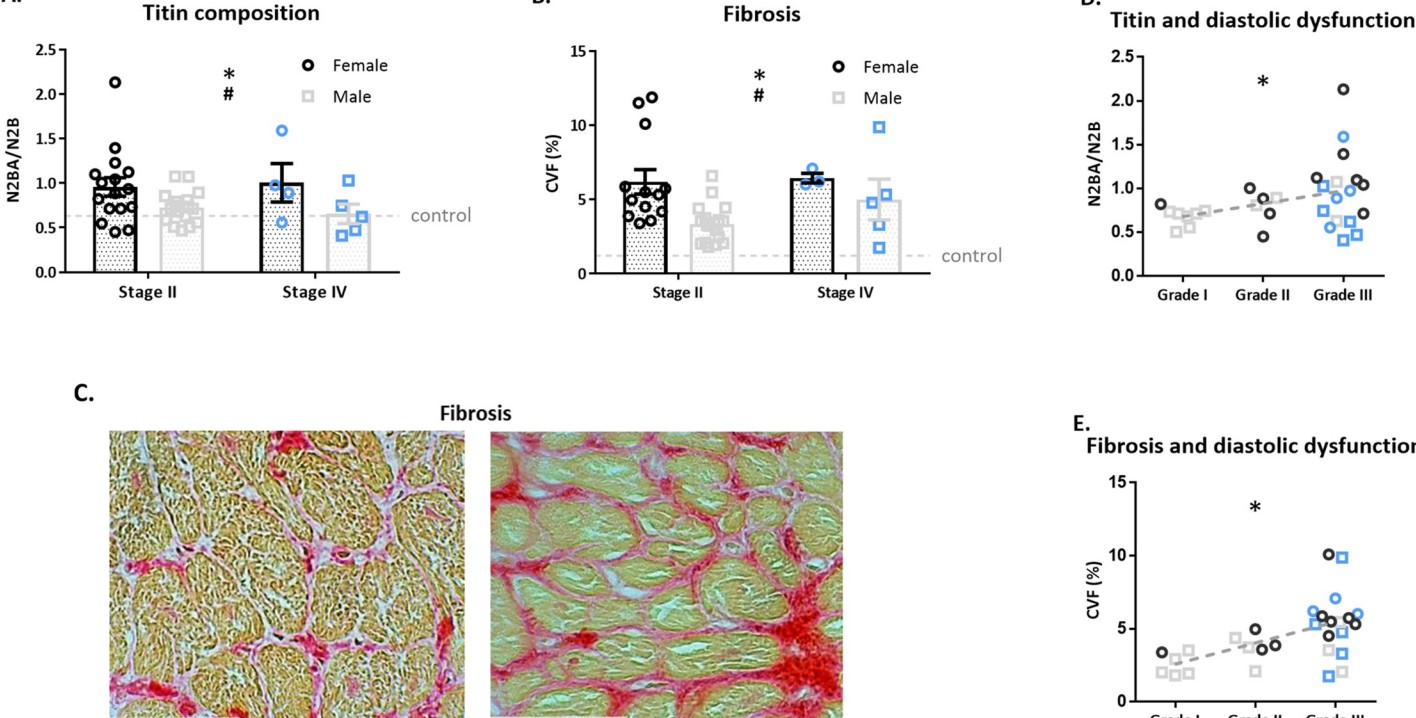

**Fig 3. Titin and Fibrosis–Female patients show more complaint titin isoform and fibrosis than male patients.** A) Diseased hearts (n = 42) show an increase in titin N2BA/N2B ratio compared to controls (p<0.05). The difference in titin composition is mainly attributed to the female patients, who show an increase in compliant titin compared to male patients (p<0.05) and compared to controls (p<0.01)(no sex-difference in N2BA/N2B ratio was present in the control group). B) Representative Picro-Sirius red stainings of HCM stage II and IV patient samples. C) The amount of fibrosis is depicted as collagen volume fraction (CVF). In comparison to controls, fibrosis is increased in the diseased hearts (stage II (n = 29) and stage IV HCM (n = 8), p<0.001. Female patients show higher levels of fibrosis compared to male patients (p<0.05). Significant correlations were found between the N2BA/N2B ratio and grade of diastolic dysfunction (p<0.05; panel E), and fibrosis and diastolic function (p<0.05; panel F). Blue dots and squares depict stage IV HCM patients. Controls used for titin isoform analysis: 1–3, 5–6, 8, 10, 13–14, 16, 20, 22, 24, 26–27. Controls used for the amount of fibrosis: 4, 8–10, 21, 23, 26, 29–30.

## Discussion

We investigated if cardiac tissue properties were more severely affected at stage IV (end-stage) than stage II of HCM, and whether these properties change in a sex-specific manner. The main findings of our study are: 1) Changes in (functional) myofilament properties of cardio-myocytes are similar in stage II and stage IV HCM; 2) A similar increase in fibrosis compared to controls is present at stage II and IV HCM; 3) Capillary density is significantly lower at stage IV compared to stage II HCM; 4) Sex-specific differences in HCM are marked by higher levels of fibrosis, a larger shift to compliant titin isoform and a lower capillary density in female compared to male patients; 5) The disease-stage specific decrease in capillary density is largely explained by the lower capillary density in men at stage IV compared to stage II. Overall, our study indicates that loss of capillary density may be a factor underlying disease progression from stage II to IV in HCM.

Age at time of operation (myectomy or heart transplantation) did not differ between the two HCM groups and is in line with previous studies.[10,24,25] HCM onset at a young age is one of the risk factors for developing stage IV HCM. Furthermore, HCM patients that progress to stage IV more often experience atrial fibrillation, mitral valve regurgitation, more symptom progression and less LVOTO.[24,26] Many stage IV patients did not have surgical myectomy

A.

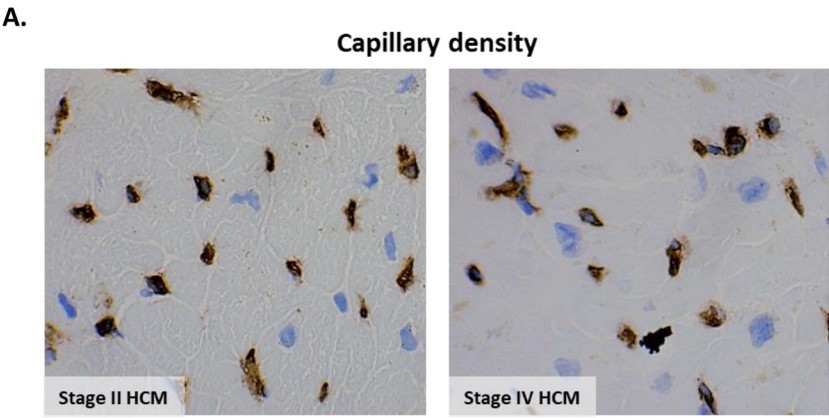

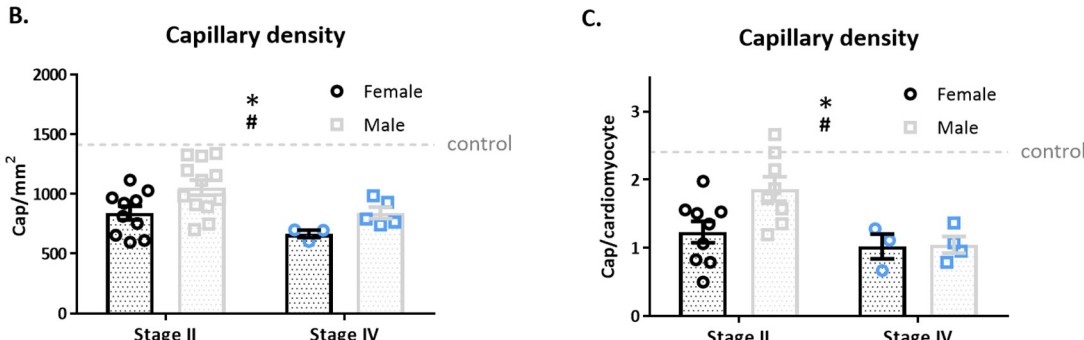

**Fig 4. Capillary density–decreased capillary density in HCM patients.** A) Representative CD31 staining of HCM stage II and IV samples are shown. B) Capillary density is depicted as the number of capillaries per mm². Capillary density is decreased in HCM stage II (n = 22) and stage IV (n = 8) compared to controls (p<0.0001). The change in capillary density is sex-dependent. Female patients show lower capillary density compared to male patients (p<0.05). C) Capillary density is depicted as the number of capillaries per cardiomyocyte. Capillary density is decreased in HCM patients compared to controls (p<0.05). Capillary density is significantly decreased in stage IV compared to stage II male patients (p<0.05). Blue dots and squares represent stage IV HCM patients. Controls used for capillary density: 8–9, 21, 23, 25–26, 29–30.

before heart transplantation, probably due to the lower frequency of LVOTO in these patients. [24] Because stage IV patients reach phase II of HCM at a younger age and progress faster, their age at heart transplantation is similar to HCM patients who undergo surgical myectomy.

As expected, cardiac remodeling was different between stage II and IV HCM patients (Table 2). Stage IV patients show less hypertrophy of the IVS and increased dimensions of both LV and left atrium.[8] Both our patient groups showed a predominance of the male sex (stage II: 56% and stage IV: 60%) which is in line with earlier research (ranging from 55–72% in end-stage HCM).[25–30] As for stage II HCM patients, based on our recent study, we have proposed that women may be diagnosed too late because the cut-off value for IVS is not corrected by body surface area.[15] The latter may partly explain the predominance of male patients in our HCM groups. Interestingly we did find a difference in drug regimen between stage II women and men. The most prescribed drugs in both genders were betablockers, calcium channel blockers or a combination of both. Men were commonly prescribed betablockers (80% in men versus 58% in women), while women were more frequently prescribed calcium channel blockers (58% versus 27%) and dual therapy (38% versus 20%).

The main difference identified between early and end-stage HCM patient groups was the larger reduction in capillary density in stage IV compared to stage II. It is known that myocardial blood flow is decreased in HCM patients, with greatest impairments seen during hyperemic circumstances.[31] The IVS and sub-endocardial layers are particularly hypo perfused. [32,33] Importantly, reduced myocardial blood flow has been associated with fibrotic areas of the myocardium.[32] Chronic exposure to ischemia, whether due to coronary occlusion or the inability to properly vascularize the hypertrophied myocardium (decreased capillary density), [5,34] leads to necrosis, massive fibrosis and eventually wall thinning.[26,35] Microvascular dysfunction in HCM patients was associated with a higher incidence of end-stage heart failure. [32,36] The lower capillary density in stage IV compared to stage II HCM patients indicates that reduced coronary perfusion plays a role in disease progression from HCM to end-stage heart failure.

We are the first to report a sex-difference in capillary density in stage II HCM patients, where woman have lower capillary density compared to men. Previous reports that analyzed capillary density have found no influence of sex but were not performed in HCM patients.[37–41] A recent study showed a correlation between microvascular density and fibrosis, in which a decrease in capillary density was correlated with an increase in fibrosis.[37] Furthermore, parameters of diastolic dysfunction were negatively associated with microvascular density.[37] Interestingly, a significant decrease in capillary density was found only in male patients between HCM stage II and IV. Capillary density in men with stage IV HCM decreased to the level of women with HCM. It has been reported that myocardial blood flow in women at rest is higher compared to men, while the flow reserve (ie. increase in myocardial blood flow between rest and stress) is lower in women.[42–44] In addition, a correlation was found between myocardial blood flow and diastolic dysfunction in women.[42] Furthermore, endothelial cell response may be impaired and contribute to HCM disease progression. Future studies are warranted to define the cellular components which underlie perturbations in myocardial blood flow during HCM disease progression.

Titin is known to modulate passive stiffness of the cardiomyocyte in an isoform-dependent manner as it functions as a molecular spring.[32] The heart consists of two isoforms, a short and stiff N2B isoform and a longer more compliant N2BA isoform. Titin-based stiffness can therefore be displayed as the N2BA/N2B isoform ratio. Our previous study[15] showed a larger shift to more compliant titin isoform in female compared to male stage II HCM patients, which coincided with a higher level of fibrosis. This is in line with the suggestion that the increased titin compliance might be an attempt to compensate for the increased fibrosis and diastolic dysfunction. Our current study shows that titin isoform composition is not altered during disease progression as a similar pattern is seen at stage IV HCM (Fig 3A). In accordance, the passive force measurement curves of both HCM groups clearly overlap (Fig 2B), indicating that passive properties of myofilaments are not altered during HCM disease progression.

All of our patients that underwent myectomy were labeled as stage II HCM. Our previous and current study indicate that tissue remodeling is more advanced in women than in men at an earlier (stage II) disease stage, i.e. more fibrosis, a change in titin isoform composition and a reduction in capillary density. Based on our observations, our HCM stage II women may actually be at a more advanced disease stage and may be in need of a more aggressive treatment.

## Clinical perspectives and relevance

### What is new?

Since no differences were observed in cardiomyocyte contractile properties between early and advanced stage of HCM, our study points to microvascular ischemia and fibrosis as main

mechanisms of disease progression rather than cardiomyocyte dysfunction. This suggests that similar molecular therapeutic targets may be at play in all phases of HCM within the myocardium, and that the different clinical profiles are mostly due to extra-cardiomyocyte variations.

## What are the clinical implications?

Analyses of coronary perfusion may be warranted in HCM patients who show a reduction in IVS during regular clinical check-ups to identify HCM patients at risk to develop end-stage heart failure.

## Translational outlook

Capillary density is only one component of microvascular dysfunction, together with loss of small vessel vasodilatory function and vessel compression due to increased wall stress. Future research could focus on evidence of ischemia and clinical research regarding small vessel function.

## Supporting information

**S1 Table. Controls.** Abbreviations: F (female); M (male).
(DOCX)

## Acknowledgments

We would like to give special thanks to Maike Schuldt who has helped with the embedding of cardiac tissue. Furthermore, we appreciate the technical contribution of Elisa Meinster, Ruud Zaremba and Wies Lommen. We would like to thank Vasco Sequeira for the design of Fig 1B.

## Author Contributions

**Conceptualization:** Louise L. A. M. Nijenkamp, Diederik W. D. Kuster, Jolanda van der Velden.

**Data curation:** Louise L. A. M. Nijenkamp, Ilse A. E. Bollen.

**Formal analysis:** Louise L. A. M. Nijenkamp.

**Investigation:** Louise L. A. M. Nijenkamp.

**Methodology:** Louise L. A. M. Nijenkamp.

**Resources:** Cris G. dos Remedios, Corrado Poggesi, Carolyn Y. Ho.

**Supervision:** Diederik W. D. Kuster, Jolanda van der Velden.

**Writing – original draft:** Louise L. A. M. Nijenkamp.

**Writing – review & editing:** Louise L. A. M. Nijenkamp, Ilse A. E. Bollen, Hans W. M. Niessen, Cris G. dos Remedios, Michelle Michels, Corrado Poggesi, Carolyn Y. Ho, Diederik W. D. Kuster, Jolanda van der Velden.

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
