## [Decision Letter · Decision Letter 0]

12 Feb 2020

PONE-D-20-01193

Sex-specific cardiac remodeling in early and advanced stages of hypertrophic cardiomyopathy

PLOS ONE

Dear Ms Nijenkamp,

Thank you for submitting your manuscript to PLOS ONE. After careful consideration, we feel that it has merit but does not fully meet PLOS ONE’s publication criteria as it currently stands. Therefore, we invite you to submit a revised version of the manuscript that addresses the points raised during the review process.

Please address the reviewers comments and pay attention to the punctuation, spelling and grammar in the text of the manuscript.

We would appreciate receiving your revised manuscript by Mar 28 2020 11:59PM. To enhance the reproducibility of your results, we recommend that if applicable you deposit your laboratory protocols in protocols.io, where a protocol can be assigned its own identifier (DOI) such that it can be cited independently in the future. For instructions see: http://journals.plos.org/plosone/s/submission-guidelines#loc-laboratory-protocols

We look forward to receiving your revised manuscript.

Kind regards,

Aldrin V. Gomes, Ph.D.

Academic Editor

PLOS ONE

Journal Requirements:

The study protocol was approved by the local Ethics Committees, and written consent was obtained.

3. We noted in your submission details that a portion of your manuscript may have been presented or published elsewhere. ["part of the stage II HCM data has been part of a manuscript which was published in Circulation Heart Failure"] Please clarify whether this publication was peer-reviewed and formally published. If this work was previously peer-reviewed and published, in the cover letter please provide the reason that this work does not constitute dual publication and should be included in the current manuscript.

Reviewers' comments:

Reviewer's Responses to Questions

**Comments to the Author**

1. Is the manuscript technically sound, and do the data support the conclusions?

Reviewer #1: Yes

Reviewer #2: Yes

2. Has the statistical analysis been performed appropriately and rigorously? 

Reviewer #1: Yes

Reviewer #2: Yes

3. Have the authors made all data underlying the findings in their manuscript fully available?

Reviewer #1: No

Reviewer #2: Yes

4. Is the manuscript presented in an intelligible fashion and written in standard English?

Reviewer #1: Yes

Reviewer #2: Yes

5. Review Comments to the Author

Reviewer #1: 1) The abstract should clearly represent the rationale and snapshot of the methods used. It must be reorganized and rewrite

2) Common punctuation, spelling and grammar mistake throughout the manuscript

3) The method section is too concise needs elaboration in terms of approaches used for the study. Additionally, vendor/manufacturer information should be reported in an eligible scientific fashion.

4) In both results and figure legends, the inference of data should be shifted to the discussion section.

5) In figure 3D, regarding diastolic dysfunction measurement, functional data/grading parameters are not provided in the manuscript. Table 1 only provides dimension-based data.

Reviewer #2: The manuscript by Nijenkamp and colleagues sought to assess cardiomyocyte contractile and cardiac muscle tissue properties from a subset of male and female HCM patients that progress to severe end-stage HCM to identify factors underlying the transition from stable stage II to progressive stable IV HCM. Overall, this is a well-executed study that specifically exploits a defined cohort of HCM patients to better dissect whether defects in contractile properties precede the structural destruction of cardiac muscle in HCM in a patient context. The authors suggest that HCM disease progression does not correlate with contractile muscle deficits, but instead titin muscle compliance, fibrosis and capillary density, which may also be impacted in a sex-specific manner. Comments are included to strengthen the manuscript.

Minor Comments:

1. It would be helpful if the authors would diagram the mutations (in Figure 1) in the context of domains of each protein as it may provide broad overview as to which mutations may be the most deleterious or pathogenic (or whether there is clustering of mutations at a specific domain).

2. It is not clear what is significant and which groups are being compared in Figure 3A and Figure 4A for significance. If possible, representative images from blots should be shown to highlight male versus female differences in each of these contexts with controls shown.

3. It would be helpful to note and discuss drug regimen (length of time on drug regimens) of HCM patients in these cohorts as it may impact readouts when assessing progression of HCM in these patients (eg., fibrosis, etc.)

4. The authors should clarify and include the contribution of titin isoform switch in their discussion to suggest its potential importance in sex differences as it is shown in their data (Figure 3A).

6. PLOS authors have the option to publish the peer review history of their article (what does this mean?). If published, this will include your full peer review and any attached files.

Reviewer #1: No

Reviewer #2: No

---

## [Author Response · Author response to Decision Letter 0]

28 Mar 2020

We have formatted the title page to meet the requirements of PLOS ONE. All Figure citations have been changed and figures have been uploaded as .tif files. Supporting Tables have now been submitted in separate files and a list of supporting information has been added to the manuscript. Citations have been formatted to Vancouver style with brackets. 

2. Thank you for including your ethics statement: The study protocol was approved by the local Ethics Committees, and written consent was obtained.

We have added the full name of the METC in brackets: 

This study was approved by the local ethics board of the Erasmus Medical Center (protocol number MEC-2010-40) and written informed consent of patients was obtained. Non-failing donor samples were acquired from the University of Sydney, Australia, with the ethical approval of the Human Research Ethics Committee (#2012/2814). 

3. We noted in your submission details that a portion of your manuscript may have been presented or published elsewhere. ["part of the stage II HCM data has been part of a manuscript which was published in Circulation Heart Failure"] Please clarify whether this publication was peer-reviewed and formally published. If this work was previously peer-reviewed and published, in the cover letter please provide the reason that this work does not constitute dual publication and should be included in the current manuscript.

As stated in our cover letter, this publication has been published in Circulation Heart Failure (2018). At first submission we have uploaded a copy of this paper. It was peer-reviewed and this work does not constitute dual publication. Some patients in the stage II HCM group have been included in both our previous study and in this manuscript. For this manuscript we have added more patients to the stage II HCM group in line with our expanding biobank. In addition to our previous study, we studied capillary density which we believe may play a central role in HCM disease progression. 

Due to our expanding biobank we have recently started to use our sample codes in published papers to give a clear overview of the samples that are included in each study. This is why we have now updated our table (Table 1) with the biobank sample codes. In the future this will increase transparency and comparison between studies. 

We have included captions for our Supporting Information files at the end of our manuscript, and have updated in-text citations to match accordingly. 

Reviewers' comments:

Reviewer's Responses to Questions

Comments to the Author

3. Have the authors made all data underlying the findings in their manuscript fully available?

Reviewer #1: No

Reviewer #2: Yes

We believe the data reviewer #1 is referring to is now displayed in total in our Supporting information files. 

 

Reviewer #1: 

1) The abstract should clearly represent the rationale and snapshot of the methods used. It must be reorganized and rewrite

As requested we have rewritten the abstract, which now includes the rational and details of the methods: 

Hypertrophic cardiomyopathy (HCM) is the most frequent genetic cardiac disease with a prevalence of 1:500 to 1:200. While most patients show obstructive HCM and a relatively stable clinical phenotype (stage II), a small group of patients progresses to end-stage HCM (stage IV) within a relatively brief period. Previous research showed sex-differences in stage II HCM with more diastolic dysfunction in female than in male patients. Moreover, female patients more often show progression to heart failure. Here we investigated if differences in functional and structural cardiac properties may underlie sex-differences in disease progression from stage II to stage IV HCM. Cardiac tissue from stage II and IV patients was obtained during myectomy (n=54) and heart transplantation (n=10), respectively. Isometric force was measured in membrane-permeabilized cardiomyocytes to define active and passive myofilament force development. Titin isoform composition was assessed using gel electrophoresis, and the amount of fibrosis and capillary density were determined with histology. In accordance with disease stage-dependent adverse cardiac remodeling end-stage failing patients showed a thinner interventricular septal wall and larger left ventricular and atrial diameters compared to stage II patients. Cardiomyocyte contractile properties and fibrosis were comparable between stage II and IV, while capillary density was significantly lower in stage IV compared to stage II. Women showed more adverse cellular remodeling compared to men at stage II, evident from more compliant titin, more fibrosis and lower capillary density. However, the disease stage-dependent reduction in capillary density was largest in men. In conclusion, the more severe cellular remodeling in female compared to male stage II patients suggests a more advanced disease stage at the time of myectomy in women. Changes in cardiomyocyte contractile properties do not explain the progression of stage II to stage IV, while reduced capillary density may underlie disease progression to end-stage heart failure.

2) Common punctuation, spelling and grammar mistake throughout the manuscript

We have subjected our manuscript to a spelling and grammar control. Furthermore we have asked a native English speaker to carefully read the paper and adjusted the text of the revised manuscript. 

3) The method section is too concise needs elaboration in terms of approaches used for the study. Additionally, vendor/manufacturer information should be reported in an eligible scientific fashion.

We have divided our methods section in subheadings and added a new paragraph: Echocardiographic measurements, including two new references. 

Further details of our methods can be found in the referenced articles in our methods section. 

Echocardiographic measurements

Echocardiographic studies were done with commercially available systems and analyzed according to the American Society of Echocardiography guidelines.[16] Maximal wall thickness, left atrial diameter (LAD), LV end-diastolic diameter (LVEDD), and LVOTO gradient were measured. LVOTO was defined as a gradient ≥ 30 mmHg at rest or during provocation. Mitral valve inflow was recorded using pulsed wave Doppler from the apical four chamber view. Mitral E and A velocity (cm/s) and deceleration time (ms) were measured. Pulsed wave tissue Doppler imaging was used to measure septal e’ velocity (cm/s). Continuous wave Doppler in the parasternal and apical four chamber was used to measure tricuspid regurgitation (TR) velocity (m/s). Echocardiographic data and medication are shown in Table 1. For the end-stage HCM group, a limited set of echocardiographic data was obtained, and not all parameters were obtained for stage II patients.

 Diastolic dysfunction was graded as follows: grade I when E/A ratio ≤ 0.8 and E peak velocity ≤ 50 cm/s; grade III when E/A ratio ≥ 2. In patients with E/A ratio ≤ 0.8 and E peak velocity > 50 cm/s or E/A ratio > 0.8 but < 2, the E/e’ ratio (>14), LADi (>24) and TR velocity (> 2.8 m/s) were used to further differentiate diastolic function. When ≥ 2 out of 3 variables were abnormal, LA pressure was elevated and grade II diastolic dysfunction was present. When 1 out of 3 variables was abnormal, grade I diastolic dysfunction was present.[17]

16. Nagueh SF, Bierig SM, Budoff MJ, Desai M, Dilsizian V, Eidem B, e.a. American Society of Echocardiography Clinical Recommendations for Multimodality Cardiovascular Imaging of Patients with Hypertrophic Cardiomyopathy. J Am Soc Echocardiogr. 2011;24(5):473–98. 

17. Nagueh SF, Smiseth OA, Appleton CP. Recommendations for the Evaluation of Left Ventricular Diastolic Function by Echocardiography: An Update from the American Society of Echocardiography and the European Association of Cardiovascular Imaging. J Am Soc Echocardiogr. 2016(29):277–314. 

Furthermore we have added a new reference to the following subheadings:

“Isometric force measurements” and “protein analyses” 

19. Bollen IAE, van der Meulen M, de Goede K, Kuster DWD, Dalinghaus M, van der Velden J. Cardiomyocyte Hypocontractility and Reduced Myofibril Density in End-Stage Pediatric Cardiomyopathy. Front Physiol [Internet]. 2017 

4) In both results and figure legends, the inference of data should be shifted to the discussion section.

We have deleted or moved all discussion from the results and figure legends to the discussion (changes are indicated in the manuscript text). 

5) In figure 3D, regarding diastolic dysfunction measurement, functional data/grading parameters are not provided in the manuscript. Table 1 only provides dimension-based data.

We have added the echocardiographic parameters used in grading diastolic dysfunction (E/A ratio, e’ velocity and TR velocity) to Table 1, and added the following text to the results section:

All stage IV patients were given grade III diastolic dysfunction, while stage II patients included 29% with grade III, 29% with grade II and 42% with grade I diastolic dysfunction. Female stage II patients show more severe diastolic dysfunction compared to male patients: 86% grade II or III diastolic dysfunction in women compared to 42% grade II in men.

Reviewer #2: The manuscript by Nijenkamp and colleagues sought to assess cardiomyocyte contractile and cardiac muscle tissue properties from a subset of male and female HCM patients that progress to severe end-stage HCM to identify factors underlying the transition from stable stage II to progressive stable IV HCM. Overall, this is a well-executed study that specifically exploits a defined cohort of HCM patients to better dissect whether defects in contractile properties precede the structural destruction of cardiac muscle in HCM in a patient context. The authors suggest that HCM disease progression does not correlate with contractile muscle deficits, but instead titin muscle compliance, fibrosis and capillary density, which may also be impacted in a sex-specific manner. Comments are included to strengthen the manuscript.

Minor Comments:

1. It would be helpful if the authors would diagram the mutations (in Figure 1) in the context of domains of each protein as it may provide broad overview as to which mutations may be the most deleterious or pathogenic (or whether there is clustering of mutations at a specific domain).

We have added a new column to Table 1 which states the domains of the mutations. Furthermore we have added a new figure (Fig 1B) that displays the structure of the proteins myosin heavy chain, myosin binding protein-C and cardiac troponin T. We have introduced blue flags to highlight the location of the mutations.

The following text has been added to the manuscript: 

Fig 1A illustrates the different gene mutations and Fig 1B shows where mutations are located in the protein. Table 1 provides an overview of the gene mutations of all patients.

We have added the following legend to our new figure:

Fig 1B. Mutation location.

Schematic of 3 main HCM sarcomere proteins: myosin heavy chain in green (Myosin), cardiac myosin-binding protein-C in purple (cMyBP-C) and cardiac troponin T in orange (cTnT). The location of the mutations is indicated with the blue circles (M). The letters N and C stand for the N-terminus and C-terminus respectively. The numbers indicate the amino acids of the sarcomere proteins. 

2. It is not clear what is significant and which groups are being compared in Figure 3A and Figure 4A for significance. If possible, representative images from blots should be shown to highlight male versus female differences in each of these contexts with controls shown.

As stated in the methods section (now under subheading: data analyses) differences to controls are indicated by an asterisk (*), sex-differences are indicated by a hashtag (#). 

Fig 3A: there is a significant difference between HCM samples and control samples; and there is a significant difference between HCM men and HCM women. This has also been stated in the figure legend: 

A) Diseased hearts (n=42) show an increase in titin N2BA/N2B ratio compared to controls (p<0.05). The difference in titin composition is mainly attributed to the female patients, who show an increase in compliant titin compared to male patients (p<0.05) and compared to controls (p<0.01)(no sex-difference in N2BA/N2B ratio was present in the control group).

Fig 4A shows CD31 staining of stage II and IV HCM patients. When capillary density is measured per mm2 (Fig 4B) there is a significant decrease in HCM patients compared to controls; and a significant lower density in HCM women compared to HCM men (which is stated in the figure legend).

When capillary density is measured per cardiomyocyte there is a significant decrease in stage IV HCM men compared to stage II HCM men (stated in the figure legend). 

3. It would be helpful to note and discuss drug regimen (length of time on drug regimens) of HCM patients in these cohorts as it may impact readouts when assessing progression of HCM in these patients (eg., fibrosis, etc.)

We have obtained the drug regimen of 73% of our study population. We were able to obtain drug regimen at the time of surgery. Unfortunately, we were not able to obtain duration of drug regimen. We have added the new information to Table 1, which shows the drug use per patient at the time of operation. 61% of all patients used betablockers and 34% used calcium channel blockers. Of all patients 23% use both betablockers and calcium channel blockers. It is hard to make a comparison between stage II and stage IV HCM because we only have the drug regimen of 1 patient in the stage IV HCM group. 

We have added the following text in the Results section:

Drug regimen was different between female and male stage II HCM patients: 80% of the men received betablockers in contrast to 58% of the women. Calcium channel blockers were prescribed more frequently to women compared to men (58% versus 27%, respectively; Table 1). Unfortunately we could only retrieve drug regimen of one of our stage IV patients.

We added the following text to the discussion:

Interestingly we did find a difference in drug regimen between stage II women and men. The most prescribed drugs in both genders were betablockers, calcium channel blockers or a combination of both. Men were commonly prescribed betablockers (80% in men versus 58% in women), while women were more frequently prescribed calcium channel blockers (58% versus 27%) and dual therapy (38% versus 20%).

4. The authors should clarify and include the contribution of titin isoform switch in their discussion to suggest its potential importance in sex differences as it is shown in their data (Figure 3A).

As requested we have added a paragraph to the discussion:

Titin is known to modulate passive stiffness of the cardiomyocyte in an isoform-dependent manner as it functions as a molecular spring.[32] The heart consists of two isoforms, a short and stiff N2B isoform and a longer more compliant N2BA isoform. Titin-based stiffness can therefore be displayed as the N2BA/N2B isoform ratio. Our previous study[15] showed a larger shift to more compliant titin isoform in female compared to male stage II HCM patients, which coincided with a higher level of fibrosis. This is in line with the suggestion that the increased titin compliance might be an attempt to compensate for the increased fibrosis and diastolic dysfunction. Our current study shows that titin isoform composition is not altered during disease progression as a similar pattern is seen at stage IV HCM (Fig 3A). In accordance, the passive force measurement curves of both HCM groups clearly overlap (Fig 2B), indicating that passive properties of myofilaments are not altered during HCM disease progression.

---

## [Decision Letter · Decision Letter 1]

15 Apr 2020

Sex-specific cardiac remodeling in early and advanced stages of hypertrophic cardiomyopathy

PONE-D-20-01193R1

Dear Dr. Nijenkamp,

We are pleased to inform you that your manuscript has been judged scientifically suitable for publication and will be formally accepted for publication once it complies with all outstanding technical requirements.

With kind regards,

Aldrin V. Gomes, Ph.D.

Academic Editor

PLOS ONE

Additional Editor Comments (optional):

Reviewers' comments:

Reviewer's Responses to Questions

**Comments to the Author**

1. If the authors have adequately addressed your comments raised in a previous round of review and you feel that this manuscript is now acceptable for publication, you may indicate that here to bypass the “Comments to the Author” section, enter your conflict of interest statement in the “Confidential to Editor” section, and submit your "Accept" recommendation.

Reviewer #1: All comments have been addressed

Reviewer #2: All comments have been addressed

2. Is the manuscript technically sound, and do the data support the conclusions?

Reviewer #1: Yes

Reviewer #2: Yes

3. Has the statistical analysis been performed appropriately and rigorously? 

Reviewer #1: Yes

Reviewer #2: Yes

4. Have the authors made all data underlying the findings in their manuscript fully available?

Reviewer #1: Yes

Reviewer #2: Yes

5. Is the manuscript presented in an intelligible fashion and written in standard English?

Reviewer #1: Yes

Reviewer #2: Yes

6. Review Comments to the Author

Reviewer #1: (No Response)

Reviewer #2: (No Response)

7. PLOS authors have the option to publish the peer review history of their article (what does this mean?). If published, this will include your full peer review and any attached files.

Reviewer #1: No

Reviewer #2: No

---

## [Editor Report · Acceptance letter]

16 Apr 2020

PONE-D-20-01193R1 

Sex-specific cardiac remodeling in early and advanced stages of hypertrophic cardiomyopathy 

Dear Dr. Nijenkamp:

I am pleased to inform you that your manuscript has been deemed suitable for publication in PLOS ONE. Congratulations! Your manuscript is now with our production department. 

With kind regards,

on behalf of

Dr. Aldrin V. Gomes 

Academic Editor

PLOS ONE